# Renal Arterial and Venous Doppler in Cardiorenal Syndrome: Pathophysiological and Clinical Insights

**DOI:** 10.3390/biomedicines12061166

**Published:** 2024-05-24

**Authors:** Roberta Barone, Vito Di Terlizzi, Giovanni Goffredo, Domenico Paparella, Natale Daniele Brunetti, Massimo Iacoviello

**Affiliations:** 1Cardiology Unit, Polyclinic University Hospital of Foggia, 71121 Foggia, Italy; barone.r90@gmail.com (R.B.); vitodt89@gmail.com (V.D.T.); natale.brunetti@unifg.it (N.D.B.); 2Department of Medical and Surgical Sciences, University of Foggia, 71121 Foggia, Italy; giovannigoffredo1993@gmail.com (G.G.); domenico.paparella@unifg.it (D.P.)

**Keywords:** cardiorenal syndrome, renal resistance index, renal doppler, renal venous flow

## Abstract

In recent decades, there has been considerable effort in investigating the clinical utility of renal Doppler measurements in both cardiovascular and renal disorders. In particular, a measure of renal arterial resistance, the renal resistive index (RRI), has been demonstrated to predict chronic kidney disease progression and acute kidney injury in different clinical settings. Furthermore, it is linked to a poorer prognosis in individuals suffering from chronic heart failure. Examining the renal venous flow through pulsed Doppler can offer additional insights into renal congestion and cardiovascular outcomes for these patients. This review seeks to summarize the existing data concerning the clinical significance of arterial and venous renal Doppler measurements across various cardiovascular and renal disease contexts.

## 1. Introduction

The usefulness of renal Doppler measurements in order to assess the renal flow was proposed several years ago by Léandre Pourcelot, who first proposed the measurement by pulsed Doppler of the renal resistive index (RRI), i.e., the difference between peak systolic and end-diastolic velocity divided by the peak systolic velocity, using arterial ultrasound Doppler waveforms [1]. Initially, the RRI was conceived as a gauge of vascular resistance across various territories but gained attention primarily for enhancing the diagnostic capabilities in the presence of urinary obstruction or renal vein thrombosis [2]. More recently, the RRI has been studied to obtain clinical findings for diagnosis and future outcomes across various cardiac and renal clinical conditions. Its clinical relevance is the result of its determinants, i.e., a complex interplay among renal and systemic vascular characteristics and hemodynamic factors [3], influenced by both parenchymal and vascular abnormalities [4].

Over the last few years, in addition to renal arterial flow measurements, the Doppler study of the renal venous flow has also shown the ability to provide further clinically relevant information about renal function [5]. In particular, the renal venous flow might offer a more precise indication of the existence of renal congestion, which could be responsible for renal function impairment and a poor prognosis.

Technological advances are significantly shaping the medical practice. In recent years, machine learning and big data have emerged as valuable tools for identifying high-risk populations and assisting physicians in clinical practice. Within this framework, it is essential to assess whether clinical and instrumental indices, such as the RRI, offer comparable utility to these new technological instruments or provide additional value to the data being analyzed [6]. Moreover, the RRI has recently been recognized as an indicator of how nephroprotective drugs may affect the kidney. Sodium–glucose cotransporter 2 inhibitors (SGLT2is) and angiotensin-converting enzyme inhibitors (ACE-Is) lead to an initial decrease in the glomerular filtration rate (GFR), the inhibition of oxidative stress markers, additional reductions in blood pressure, and overall peripheral resistance. Notably, long-term gliflozin preserves renal perfusion and lowers the vascular resistance [7].

This review offers perspectives on the clinical utility of performing arterial and venous renal Doppler measurements in conditions affecting both the heart and the kidneys. In order to achieve the goal of our research, we conducted searches in electronic databases such as PubMed, using specific keywords or locating articles related to renal resistance index and cardiovascular diseases, arterial renal Doppler, venous renal Doppler and congestion, and cardiorenal syndrome. Following the primary keyword searches in databases and adhering to the predetermined criteria for including and excluding literature, we selected studies based on the incidence rate of listening issues and the frequency of pure-tone hearing thresholds.

## 2. Methodological Aspects of Renal Doppler Evaluation

In the traditional method, the Doppler procedure starts with the patient lying on the back. A low-frequency probe (2.5–5.0 MHz) visualizes the kidney and its associated vessels. Imaging is typically conducted through two primary approaches: the anterior abdominal wall or the flank. Generally, the anterior approach is preferred for assessing the kidney parenchyma and the renal arteries, while the flank approach may be utilized to extensively observe both renal arteries and veins from the vessel to the hilum. The patient remains supine while the probe is maneuvered inferiorly and superiorly to locate the renal arteries and any additional arteries. Both B-mode and color Doppler are utilized to aid in the identification [8]. Renal Doppler imaging can also be conveniently conducted after a standard echocardiography using the same 4 MHz probe (Figure 1). In this case, with the patient seated, a posterior approach to the kidney is employed. After obtaining a clear two-dimensional image of the renal parenchyma, a color Doppler evaluation of the intrarenal vascular blood flow is performed. For the Doppler measurements, the segmental or interlobar arteries are sampled because they provide the best Doppler signal for the flow quantity and the correct angle. The course of the segmental arteries of either the right or the left kidney is observed using color Doppler flow imaging. Subsequently, pulsed Doppler detection is conducted at the mid-tract level of the most clearly visualized artery. Special attention is given to guarantee that the ultrasonic beam is correctly aligned. It is recommended to take an average of 2–3 measurements for both peak systolic velocity and end-diastolic velocity to calculate the RRI.

The peak systolic velocity and end-diastolic velocity measurements are utilized in Peurcelot’s formula to calculate the RRI. This formula involves subtracting the end-diastolic velocity from the peak systolic velocity and then dividing the result by the peak systolic velocity [1]. Typically, the mean value of three measurements for each kidney is used. A normal RRI value is generally considered to be 0.60 ± 0.01 (mean ± standard deviation), with most authors regarding the value of 0.70 as the upper normal threshold [9].

In order to analyze the renal venous flow, the course of the right or left kidney veins is identified using color Doppler flow imaging. Then, pulsed Doppler imaging is performed at the mid-segment level of the most distinctly visible vein in order to register the pattern. Venous Doppler recordings are made at end expiration [10,11,12]. Various venous patterns exist, as shown in Figure 2.

The typical pattern is defined by a slowing of the presystolic flow velocity and biphasic patterns without interruption of the tele-diastolic flow. Additionally, there may be a continuous flow with minimal variations. Finally, different intermittent renal venous flows can be present. A minimally intermittent pattern displays a short tele-diastolic interruption of the forward flow with a reversal flow. More pronounced intermittent flows show a biphasic interruption or a reversal within the same cardiac cycle until one forward and one reversal wave flow appear, which is denoted as a monophasic intermittent pattern. The published studies classified these five possible patterns: types A and B for continuous flows, C for mild intermittent flows, D for biphasic flows, and E for monophasic intermittent flows [13].

## 3. Arterial and Venous Renal Doppler: The Pathophysiological Background

The RRI is a measure that could reflect the complex pathophysiological background underlying changes in arterial renal resistances. As shown in Figure 3, both dynamic and structural changes within intrarenal vessels could be responsible for elevated RRI values (≥0.70) [14].

An elevated RRI could reflect a physiological response to increased flow and/or pulsatility resulting from stiffer aorta and vascular system [15]. From a pathophysiological point of view, the correlation between RRI and blood pressure levels can be explained mathematically based on the formula that defines the index. An elevation in pulse pressure increases the difference between the peak of renal velocity and the end-diastolic velocity [16]. Moreover, in a series of patients with essential hypertension, a significant and independent association between RRI and central pulse pressure was observed [17]. In this setting, it has been hypothesized that the exposure of small renal arteries to a high blood flow, coupled with an elevated pulsatile pressure, could contribute to increasing the RRI, thus potentially affecting the microvascular bed [17]. Moreover, a chronic elevation in blood pressure levels prompts the arterial remodeling, leading to increased arterial stiffness, higher renal vascular resistance, and increased RRI [18].

In diabetic patients, an increase in RRI could result from both macrovascular and microvascular abnormalities. In the absence of microvascular lesions, renal macrovascular lesions may partially account for the varying relationships observed between RRI, urinary albumin excretion, and creatinine clearance [19]. Non-albuminuric renal insufficiency might be linked to other pathogenetic disorders, such as ischemic nephropathy, rather than classic diabetic nephropathy [20]. Renal macrovascular disease damage could also lead to renal hypoperfusion and reduced kidney blood flow. Consequently, within the microvascular bed, vessel vasoconstriction tries to maintain an adequate organ perfusion, resulting in increased resistance and elevated RRI values [21]. However, the precise role and extent of glomerular involvement in the elevation of the RRI remain unclear [22]. However, it should be highlighted that the afferent and efferent arterioles account for the main part of renal resistances, thus making likely their influence on a Doppler measure of renal resistances. This is even more likely considering that the activation of the sympathetic and renin–angiotensin systems can modify the renal circulation mainly regulating the glomerular arterioles’ tone and, thus, the RRI values [14]. Finally, prolonged vasoconstriction can result in vascular rarefaction, characterized by a reduction in the number of functional vessels, leading to tissue ischemia, endothelial dysfunction, and ultimately fibrosis and vascular rarefaction [23].

In the pathophysiological background underlying an increased RRI, increased intra-abdominal pressure and central venous pressure (CVP) should also be considered [24,25]. An increased CVP causes renal congestion and leads to heightened pressures in the efferent arterioles and glomerular capillaries, thereby reducing net filtration pressure and GFR [26]. It happens because the casing that encloses the kidney is firm and expandable. For this reason, the CVP can play a distinct role in raising the RRI, which suggests a potential overall elevation in intrarenal arterial resistance. However, a high CVP should be considered not only in the pathophysiology underlying the RRI values but also in the abnormalities of the renal venous flow. When hemodynamic shifts occur in the systemic venous circulation, resulting in elevated venous pressure, abnormalities in venous Doppler profiles can be observed at various locations [5]. For this reason, detecting abnormal Doppler flow patterns helps diagnose a clinically significant systemic venous congestion. Moreover, organ congestion due to venous hypertension may contribute to organ injury in patients with critical illnesses, congestive heart failure, and chronic kidney disease across different clinical scenarios [14,27].

## 4. The Renal Resistance Index in Cardiorenal Settings

In clinical practice, the RRI is a valuable tool across various medical settings and patient contexts. Numerous studies suggest the relevance of this index in different clinical conditions, such as hypertension, diabetes, heart failure, contrast medium- or cardiac surgery-induced acute kidney injury (AKI), sepsis, COVID-19, and prior renal replacement therapy as shown in Table 1.

### 4.1. RRI and Cardiorenal Syndrome in Essential Hypertension

The relationship between arterial renal flow abnormalities and essential hypertension has been evaluated in several studies. In hypertensive patients, left ventricular dilatation and subclinical renal damage (albuminuria) were associated with early intrarenal vascular alterations. In particular, the ratio of ultrasound renal volume to RRI was associated with nephroangiosclerosis, increased renovascular stiffness [28], and a significant risk of renal disorders [29,30].

Elevated RRIs were also associated with an increased risk of cardiovascular morbidity, mortality, and renal outcomes in both essential hypertensive and chronic kidney disease (CKD) [31,32,33]. The association between a high RRI and cardiovascular and renal outcomes was significant even in the subgroup with eGFR < 60 mL/min per 1.73 m^2^ [31]. Moreover, high RRI and low estimated glomerular filtration rate (eGFR) strongly predicted adverse outcomes. In particular, RRI ≥ 0.70 was the identified cutoff value significantly linked to increased all-cause mortality [18]. Interestingly, the risk for all-cause mortality was adjusted for systolic and diastolic blood pressure, suggesting that the RRI holds prognostic value beyond the pulse pressure [34].

From a therapeutic point of view, evaluating the RRI may assist clinicians in selecting appropriate drug therapies. Indeed, an elevated RRI may be a signal of increased intrarenal stiffness, prompting caution in titrating renin–angiotensin system inhibitors to prevent renal function decline, particularly in CKD patients, diabetics, or the elderly. Conversely, previous studies have indicated that specific antihypertensive agents impact the RRI. Renin–angiotensin system inhibitors such as valsartan and lisinopril can enhance the renal function by reducing the renal vascular resistance, particularly in patients with essential hypertension and in those exhibiting microalbuminuria [35].

### 4.2. RRI and Cardiorenal Syndrome in Diabetic Patients

In diabetic patients, a close association exists between the vascular system and the kidney, likely the primary driver of diabetic nephropathy (DN) due to complications affecting both the micro- (intrarenal) and the macrovascular beds and atherosclerosis of intra- and extrarenal arteries. DN typically manifests initially with glomerular hyperfiltration, followed by the onset of microalbuminuria, progressing to macroalbuminuria, ultimately culminating in a decline in GFR. Microalbuminuria and serum creatinine have been regarded as predictors of DN. However, their predictive value is limited, as their levels rise significantly when renal damage has occurred. Over the last years, alternative biomarkers have been studied to identify early signs of DN. Galectin-3 has been linked to glomerular injury, but high levels have also been observed in the general population, including individuals without CKD. Growth differentiation factor-15 (GDF15) has been implicated in the progression of kidney dysfunction, mortality, and morbidity in patients with type-1 diabetes mellitus and altered kidney function. Nevertheless, there is currently insufficient evidence regarding the utility of galectin-3 and GDF-15 as biomarkers of DN [36].

The early identification of diabetes and its vascular complications is crucial for mitigating the impact of this disease. In this setting, the RRI could be useful for assessing intrarenal hemodynamic abnormalities due to micro- and macrovascular complications [37]. A high renal resistive index (RRI) is linked to significant arteriosclerosis, which might elucidate its connection with cardiovascular risk [4]. Reno-vascular stiffness may also contribute to the condition in individuals affected by diabetes mellitus (DM) [38]. Diabetic subjects present higher RRI values (mean, 0.72) compared to non-diabetic individuals (mean, 0.65). The RRI could serve as an early indicator of such damage, surpassing the predictive capability of eGFR [37]. Moreover, it progressively increases with a worsening renal function (WRF) [39].

### 4.3. RRI in Heart Failure Patients

In patients affected by HF, there is a high prevalence of CKD. Moreover, these two conditions can be responsible for a worse outcome [14,40]. This presents the typical characteristics of the cardiorenal syndrome [41,42]. In this setting, the RRI is a useful parameter to better characterize patients prone to cardiorenal syndrome progression. While the clinical diagnosis of heart failure relies on symptoms and physical signs of fluid overload, the sensitivity and specificity of these parameters are limited, often causing diagnostic delays and uncertainties. In recent decades, novel ultrasound techniques have emerged to detect elevated intracardiac pressures and fluid overload with greater sensitivity and specificity. These advancements enable an earlier and more accurate diagnosis, as well as more effective treatment strategies.

Studies have explored the use of kidney and venous system imaging to detect congestion, offering improved prediction of heart failure decompensation [43]. However, these methods may not fully assess the functional residual capacity of the kidneys, even in cases of compensated heart failure. Consequently, the advantage of the renal resistive index diverges slightly from that of venous congestion imaging, as it reflects the degree of congestion and renal exertion to sustain diuresis.

The RRI is associated with a worse outcome in patients affected by heart failure with preserved (HFpEF) and reduced (HFrEF) ejection fraction [24,32]. In a series of 90 HFpEF patients and 90 age- and sex-matched hypertensive patients, the mean RRI was independently associated with HFpEF and was an independent predictor of poor outcome [32]. Similarly, the RRI was associated with events in univariate and multivariate Cox regression analysis in 250 HFrEF outpatients in a stable clinical condition [24]. Moreover, the RRI may help identify CHF patients who are susceptible to worse renal outcomes, such as those at risk of developing diuretic resistance (defined by the necessity for higher diuretic doses) and WRF, throughout mid-to-long-term follow-up [44,45]. On the other hand, treatments able to improve the cardiac function, such as sacubitril/valsartan, could also induce an improvement in the RRI [46]. These data emphasize the valuable role of the RRI in evaluating the renal hemodynamic status and forecasting the risk of heart failure progression, renal decline, and adverse events in patients affected by heart failure.

In recent years, various machine learning models have been extensively tested for the early detection of heart failure (HF) patients, a subject investigated, notably, by Chen et al. Among these models, recurrent neural networks (RNNs) have proven notably effective in predicting future heart failure diagnoses, exceptionally when trained on ample datasets [43]. There needs to be more comparative data regarding these methods and the RRI in predicting CS. Nevertheless, including RRI data within a machine learning model could enhance the accuracy of heart failure diagnosis in this context [47].

### 4.4. RRI and Contrast-Mediated AKI

Contrast medium-induced AKI represents a clinically relevant condition among patients undergoing radiologic examinations such as coronary angiography because associated with an increased morbidity [48]. According to the KDIGO definition, AKI affects one out of five adults globally (21.6%). Specifically, contrast-induced AKI is characterized by a rise in serum creatinine of 0.3 mg/dL or a creatinine increase to ≥1.5 times the baseline value within 3 to 5 days after contrast exposure. Data from the American National Cardiovascular Data Registry’s Cath-PCI registry indicated an overall incidence of contrast-induced AKI of 7.1%, with 0.3% of the patients necessitating dialysis [49]. Various formulations of contrast media have demonstrated direct and indirect cytotoxic effects [50,51]. When exposed to contrast media, endothelial cells undergo significant damage or apoptosis. The release of free iodine from the contrast during procedures directly affects the endothelial cells, while the contrast media also exhibit cytotoxic effects on the nephron tubules. This dual assault results in oxidative stress and the generation of free radicals and reactive oxygen species (ROS), which deplete nitric oxide (NO) and undermine its vasodilatory properties [52].

The serum creatinine levels have been considered a gold standard for the diagnosis of AKI; however, they may be affected by age, sex, muscle mass, and medications, and abnormal levels may not be detectable until the patients suffer a significant loss of kidney function. The serum creatinine levels have traditionally been used as the primary measure for diagnosing AKI; however, several factors can influence their accuracy [53]. Additionally, abnormal levels may not become apparent until a significant decline in kidney function occurs. Recently, new biomarkers like NGAL (neutrophil gelatinase-associated lipocalin), cystatin C, and KIM-1 (kidney injury molecule-1) have emerged as indicators of renal tubular injury. Even if these biomarkers can enhance clinical diagnosis, their sensitivity, specificity, and reliability are currently insufficient for their widespread clinical application [54].

Recently, we demonstrated that the RRI helps to identify patients at high risk of WRF after contrast medium administration for coronary angiography [45]. This association seems to result from the above-mentioned mechanisms underlying a high RRI [55]. This is confirmed by the fact that in patients at risk of CI-AKI undergoing cardiac catheterization, a higher post-procedural RRI predicted contrast-induced AKI with high sensitivity and specificity [56].

### 4.5. RRI and AKI after Cardiac Surgery

The complex interplay between cardiovascular and renal diseases often manifests bidirectionally and may influence their progression, particularly among patients undergoing cardiac surgery. The incidence of WRF in these cases ranges from 5% to 42% [57], and its occurrence is linked to a more severe short- and long-term prognosis, irrespective of other risk factors. However, there is limited research on predictors of WRF following cardiac surgery.

Cardiac surgery-associated AKI (CSA-AKI) is a broad term encompassing various pathologies. The pathophysiology of AKI after cardiac surgery remains incompletely understood, involving multiple factors such as hypoperfusion, ischemia–reperfusion injury, neurohumoral activation, inflammation, oxidative stress, nephrotoxins, and mechanical factors. These pathways may operate preoperatively, intraoperatively, and postoperatively or at multiple times during the surgical process, with varying degrees of impact on individual patients [58].

Bossard et al. explored if the RRI could anticipate the development of AKI in patients who underwent cardiac surgery. Their study revealed that the RRI demonstrated notable sensitivity and specificity in predicting early AKI after cardiac surgery and was strictly associated with the degree of renal impairment, surpassing traditional serum creatinine-based indicators, which showed delayed elevation following the operation [59]. In this context, assessing the RRI could be valuable for identifying patients at higher risk of CRS progression.

### 4.6. RRI in Sepsis and COVID-19

AKI affects approximately 25% of sepsis cases in the intensive care unit, a figure that rises to 50% in septic shock situations [60,61]. The underlying mechanisms are intricate, probably based on interactions between renal hemodynamic alterations that occurred during sepsis and inflammation [62]. Vascular dysfunction likely plays a central role in these underlying mechanisms [63,64,65]. Additionally, acute tubular necrosis resulting from hemodynamic instability and injury induced by inflammatory cytokines and other active mediators may contribute to the condition. Therefore, the RRI could enable the noninvasive exploration of renal hemodynamics and predict renal dysfunction.

In this setting, sepsis probably alters the RRI through multiple pathways including the mean arterial pressure, the presence of severe AKI, the duration of the intensive care unit stay, and a medical history of diabetes and/or hypertension. The association between RRI and AKI in septic patients may be partly due to inflammatory cytokines causing hemodynamic alterations [66]. During COVID-19, certain individuals may develop severe complications such as viral pneumonia, multiorgan failure including respiratory failure due to acute respiratory distress syndrome (ARDS), myocarditis, advanced renal failure, and even death. The occurrence of acute kidney injury (AKI) among COVID-19 patients ranges from 3% to 15%, but in cases requiring intensive care, this rate significantly rises from 15% to 50%. AKI has been identified as an independent risk factor for mortality in individuals with COVID-19. Moreover, COVID-19 is linked to an elevated frequency of albuminuria and hematuria. Since AKI independently contributes to increased hospital mortality among COVID-19 patients, a novel marker (other than serum creatinine) is needed in order to prevent renal damage [67]. Among patients with COVID-19, despite the serum creatinine levels remaining within the normal range, RRI values greater than 0.70 mostly occurred without a concurrent rise in AKI incidence. This suggests that RRI elevation occurs independently of renal dysfunction, probably as a response to systemic inflammation—a manifestation of the kidney’s response to the plasma conditions and the inflammatory state [68].

### 4.7. RRI and Renal Transplantation

Kidney transplantation stands as the primary treatment modality for individuals with end-stage kidney disease, offering superior outcomes in terms of morbidity, mortality, quality of life, and cost-effectiveness compared to dialysis. Concerns regarding graft function and survival after kidney transplant have underscored the importance of identifying risk factors and variables predictive of graft success. Of particular utility are factors that can be assessed early post-transplantation and may serve as predictors of long-term graft survival.

Doppler ultrasound is the primary imaging modality for assessing kidney transplant vasculature. It is a non-invasive method for monitoring the kidney graft during the immediate postoperative phase, facilitating the detection of macrovascular abnormalities and microvascular changes in intrarenal arterial blood flow. The RRI has been identified as the optimal ultrasound parameter for evaluating kidney allograft status and dysfunction. The RRI measured within the first 48 h after kidney transplantation, during intensive care unit stay, can predict short- and long-term outcomes for patients and grafts. Patients experiencing delayed graft function exhibited higher RRI levels than those who did not experience it [69,70].

**Table 1 biomedicines-12-01166-t001:** The main studies evaluating the relevance of the renal resistance index in different clinical settings.

	No. of Patients	Year	Results
Hypertension			
Leoncini G. et al. [35]	32	2002	Greater reduction in RRI with Lisinopril (0.61 ± 0.02 to 0.56 ± 0.04) than with nifedipine (0.61 ± 0.01 to 0.59 ± 0.02)
Doi Y. et al. [31]	288	2012	Higher RRI (male ≥ 0.73, female 0.72) was associated with poorer cardiovascular and renal outcomes (HR at multivariate Cox regression analysis 9.58, 95% CI, 3.26–32.89; *p* < 0.01)
Viazzi F. et al. [29]	231	2013	Patients with new-onset diabetes have low RV/RI (HR 1.04, 95%CI 1.02–1.07, *p* = 0.0012). RV/RI is inversely related to eGFR (r = 0.17, *p* = 0.0133)
**Diabetes**			
Geraci G. et al. [38]	264	2015	RRI was positively correlated with intima–media thickness in the whole population (r = 0.43; *p* < 0.001) and in subgroups with (r = 0.42; *p* < 0.001) and without (r = 0.39; *p* < 0.001) CKD
Theertha K. C. et al. [37]	114	2023	Diabetic patients showed higher RRI values than non-diabetic patients (mean 0.72 vs. mean 0.65) (*p* = 0.001). RRI was correlated with serum creatinine levels (*p* = 0.001) and eGFR (*p* = 0.001). A cut-off of 0.70 was used to define impaired RRI
**Heart failure**			
Ennezat P. V. et al. [32]	90	2011	In HFpEF patients, elevated RRI was associated with a worse outcome (HR = 1.06; 95% CI: 1.01–1.10, *p* = 0.007). RRI cut-off: 82%
Ciccone M. M. et al. [24]	250	2014	In CHF patients, RRI was associated with events at univariate (HR 1.14; 95% CI 1.09–1.19; *p* < 0.001; C-index = 0.77) as well as at multivariate Cox regression analysis (HR 1.08; 95% CI 1.02–1.13; *p* = 0.004; C-index = 0.86). RRI best cut-off for survival 75%
Iacoviello M. et al. [44]	250	2015	RRI was associated with baseline diuretic dose at univariate (OR 1.39; 95% CI: 1.233–1.58; *p* < 0.001) and multivariate analysis (OR 1.27; 95% CI: 1.09–1.49; *p*: 0.002). The increase at 1 year in diuretic dose was associated with RRI (OR: 1.37; 95% CI: 1.19–1.57; *p* < 0.001). RRI predictive cut-off 70%
Iacoviello M. et al. [45]	266	2016	RRI was associated with worsening renal function at univariate (OR: 1.13; 95% CI: 1.07–1.20) and forward stepwise multivariate logistic regression analysis (OR: 1.09; 95% CI: 1.03–1.16; *p* = 0.005). RRI predictive cut-off 70%
**Contrast-induced AKI**			
Tawfik M. et al. [56]	100	2019	Pre-procedural RRI was higher in patients developing CI-AKI (0.71 ± 0.01 vs. 0.61 ± 0.4, *p* < 0.001). Post-procedural RRI > 0.744 predicted CI-AΚI with a sensitivity of 94% and a specificity of 92%.
Barone R. et al. [55]	148	2023	RRI was independently associated with worsening renal failure (OR: 1.22; 95% CI: 1.09–1.36; *p* = 0.001) RRI predictive cut-off 70%
**Post cardiac surgery AKI**			
Bossard G. et al. [59]	65	2011	RRI was increased in patients with AKI when compared to patients without AKI (0.79 ± 0.08 vs. 0.68 ± 0.06, respectively, *p* < 0.001). RRI > 0.74 predicted delayed AKI with 85% sensitivity and 94% specificity
**Sepsis**			
Beloncle F. et al. [66]	65	2019	RRI was higher in a group of patients with more severe AKI (0.73, interquartile range [0.67; 0.78], versus 0.67 [0.59; 0.72], *p* = 0.001)
**Renal Transplantation**			
Tirtayasa P. M. W. et al. [70]	1802	2019	Meta-analysis of nine studies. Higher RRI was associated with a higher risk of delayed graft dysfunction after transplantation (RR 2.04; 95% CI: 1.72–2.41, *p* < 0.00001). The long-term survival was higher in patients with low RRI (RR 0.82; 95% CI: 0.72–0.93, *p* < 0.002)
Bogaert S. et al. [69]	478	2022	The RRI routinely measured <48 h post-transplant is an independent predictor of short-term kidney function. RRI ≥ 0% was associated with a higher mortality.

The studies in the table are categorized by their field of application, highlighted in bold. AKI: acute kidney injury; CI-AKI: contrast-induced acute kidney injury; CI: confidence interval; CKD: chronic kidney disease; eGFR: estimated glomerular filtration rate; HFpEF: heart failure with preserved ejection fraction; OR: odds ratio; RR: relative risk; RRI: renal resistance index.

## 5. Renal Venous Doppler: Clinical Aspects

Renal Doppler ultrasonography may help to complete the assessment of renal hemodynamics by evaluating the renal venous blood flow. As for arterial Doppler ultrasonography, a low-frequency probe (2.5–5.0 MHz) visualizes the kidney and its vessels. Images can be taken through two primary approaches, i.e., with the patient in a sitting position or in a lying position, placing the probe on the flank on the right posterior axillary line. Doppler waveforms are obtained on interlobar and segmental renal veins. Because the venous signal is very dependent on the respiratory phase, the Doppler spectra are obtained at the end of expiration. The Doppler waveforms of intrarenal veins are classified as in Figure 1 (from a continuous to an intermittent pattern). The arterial flow will show a positive tracing, while the venous flow will show a negative tracing. In clinical practice, the CVP estimated through the inferior vena cava diameter measurement is the standard echographic parameter used to detect renal congestion in heart failure patients. However, when hemodynamic shifts occur in the systemic venous circulation, resulting in elevated venous pressure, abnormalities in venous Doppler profiles can be observed at various locations. Detecting abnormal Doppler flow patterns helps diagnose clinically significant systemic venous congestion [14,27,71].

### 5.1. Renal Venous Flow in Chronic Heart Failure

In chronic heart failure patients, the presence of an intrarenal intermittent venous flow is linked to elevated CVP, renal congestion, and a poorer prognosis [5,12,13,72]. In particular, not only the patterns characterized by a severe intermittent flow but also a minimal intermittent pattern was independently associated with an increased risk of heart failure progression [13]. Arterial and venous renal Doppler provides distinct and complementary information: arterial Doppler/RRI reflects vascular and parenchymal renal abnormalities and neurohormonal hyperactivity, while venous patterns more accurately reflect renal congestion [71].

The evaluation of intrarenal flow patterns through renal Doppler ultrasonography has garnered attention in patients with heart failure. A discontinuous flow has been proposed to correlate with congestion and diminished diuretic efficacy. The simultaneous recording of pulsed Doppler waveforms from interlobar arteries and veins enables the rapid assessment of a patient’s filling status and facilitates the monitoring of the diuretic response [72].

Indeed, a physical examination alone to assess the fluid status is obsolete, limiting clinical decision-making effectiveness. Ultrasonography and the evaluation of venous Doppler flow patterns serve as valuable, non-invasive, bedside diagnostic tools for assessing the fluid status and managing venous congestion. These discoveries indicate the practical value of renal venous Doppler assessments to gain deeper insights into cardiorenal syndrome. Incorporating renal venous Doppler assessments into routine clinical practice could be valuable, allowing for the customization of treatment strategies accordingly.

### 5.2. Renal Venous Flow and AKI

Hermansen et al. explored the correlation between Doppler signals indicating renal perfusion and AKI onset after cardiac surgery. An irregular renal venous flow pattern detected on the initial postoperative day and an increased portal vein pulsatility fraction were linked to a heightened risk of severe AKI development [73].

Among individuals with acute coronary syndrome (ACS), the presence or absence of systemic congestion (VExUS 0/≥1) was associated with the occurrence of AKI [74]. Finally, in cardiac surgery patients, in order to determine their predictive capacity for AKI, different indicators of venous congestion such as CVP, inferior vena cava (IVC) diameter, portal pulsatility fraction (PPF), hepatic vein flow pattern (HVF), intrarenal venous flow pattern (IRVF), and VExUS were assessed. The results showed that venous congestion is linked to AKI following cardiac surgery, although it may not always correlate with the need for continuous renal replacement therapy (CRRT) [75].

### 5.3. Renal Venous Flow in Intensive Care Setting

An accurate assessment of the fluid status is crucial for effectively treating patients in the emergency department. Traditionally, clinicians have relied on examining the inferior vena cava (IVC) and detecting pulmonary B-lines to gauge fluid status and responsiveness [76].

The venous excess ultrasonography score (VExUS) offers a comprehensive approach. It involves a four-step process that not only checks for congestion in the IVC but also evaluates the severity of congestion in the liver, gut, and kidneys. This protocol equips emergency physicians with a tool to tailor fluid management strategies for their patients, including decisions on fluid administration, diuretic use, and vasopressor therapy, and could be used as part of Protocols for Point-of-Care Ultrasound (POCUS) estimation [77].

The assessment begins by examining the diameter of the inferior vena cava (IVC). If the IVC diameter measures less than 2 cm, congestion is absent, and further evaluation is unnecessary (VExUS 0). However, if the diameter exceeds 2 cm, congestion is present, and further investigation to determine its severity is needed. Next, the hepatic veins are assessed. The pulsed-wave Doppler is positioned proximal to the hepatic vein entry into the IVC. The normal hepatic vein flow exhibits a trace pattern with three waves, i.e., a small retrograde A wave, followed by anterograde S and D waves. In cases of venous congestion, the S wave’s magnitude diminishes. Following the hepatic vein assessment, attention turns to the portal vein, identified by its thick and hyperechoic walls. In normal conditions, the portal system exhibits a constant monophasic flow with minimal variation. However, as venous congestion increases, the flow becomes pulsatile. The final step involves evaluating the renal vein, following the same methodology described earlier. Normal renal veins display an uninterrupted monophasic flow or a slightly pulsatile flow, but as venous congestion increases, the intrarenal venous flow shows gradual changes, from a biphasic pattern in moderate congestion to a monophasic pattern in severe congestion [78].

The VExUS categorizes congestion into grades 0–3, as shown in Table 2.

## 6. Conclusions

Renal arterial and venous Doppler provides clinically relevant information in different clinical settings. In both chronic and acute scenarios, assessing the condition of renal arteries and veins can provide clinicians with valuable insights into patients’ fluid status, potentially guiding more effective therapeutic interventions. In particular, the RRI evaluation can help better characterize the presence of renal dysfunction and predict its acute and chronic worsening. For this reason, it is also a parameter independently associated with a poor outcome. In addition to the RRI, the evaluation of the renal venous flow provides information about renal congestion, thus representing a further prognostic parameter. A multiparametric and comprehensive evaluation can effectively define the optimal approach for each patient, aiding in monitoring therapy effectiveness and assessing the fluid status in cases of chronic heart failure.

Future studies should clarify if the information provided by renal Doppler could be useful to tailor therapeutic approaches for patients in order to reduce the risk of cardiorenal syndrome progression.

## Figures and Tables

**Figure 1 biomedicines-12-01166-f001:**
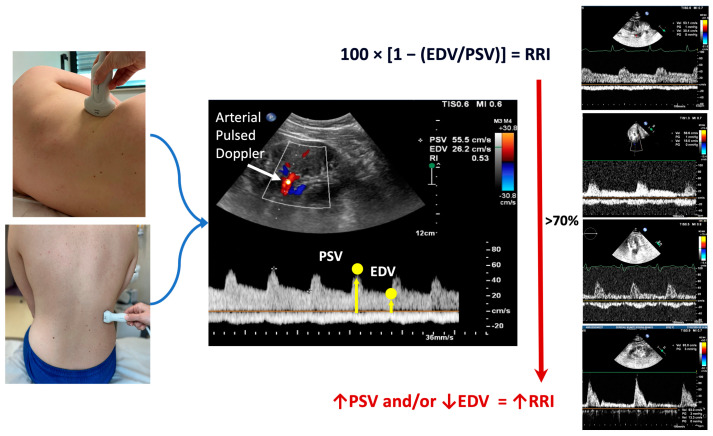
On the left of the figure, the possible approaches to conduct renal Doppler imaging are shown, i.e., the flank approach (top) in a sitting position (bottom). In the middle, the calculation of the RRI is illustrated. Finally, on the right of the figure, examples of the different renal arterial (top of the figure) and venous (bottom of the figure) Doppler flows are shown. The increase in the renal resistance index is related to the increase in the difference between peak systolic and end-diastolic velocity. The red arrow ↑ indicates the increase of the parameter; the red arrow ↓ indicates the reduction of the parameter. EDV: end-diastolic velocity; PSV: peak systolic velocity; RRI: renal resistance index.

**Figure 2 biomedicines-12-01166-f002:**
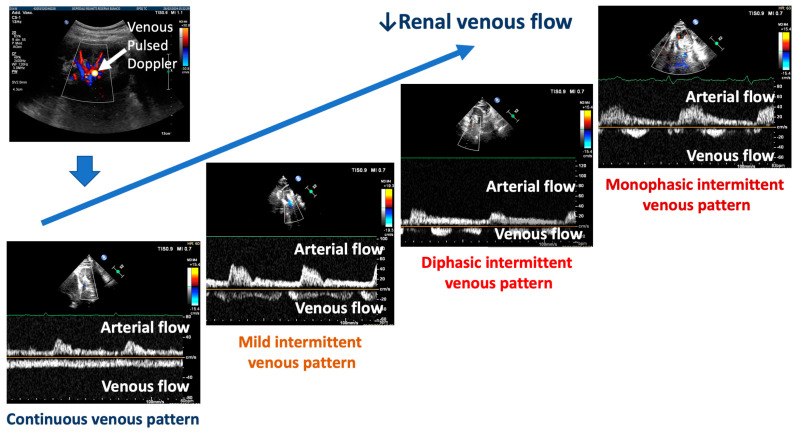
The different patterns associated with the decrease in renal venous flow are shown. Fluid overload and the increase in central venous pressure are marked by an intermittent pattern ranging from mild (mild intermittent pattern) to severe (diphasic–monophasic pattern). The red ↓ indicates the reduction of the renal venous flow.

**Figure 3 biomedicines-12-01166-f003:**
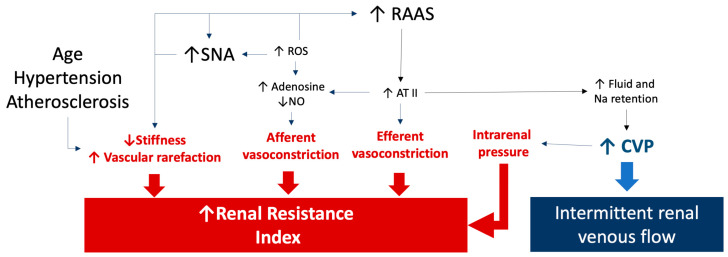
The figure summarizes the pathophysiological factors influencing the renal resistance index and renal venous Doppler results. An increase in renal resistances could also be related to an increased tone of the renal arteries as well as to a reduced stiffness caused by aging, hypertension, and atherosclerosis. Sympathetic and renin angiotensin system activation as well as oxidative stress are responsible for increased afferent and efferent arteriole resistances. Finally, a renal venous intermittent flow is related to an increase in central venous pressure, which could be also responsible for an increased resistance index due to a rise in intrarenal pressures. See the text for details and references. The arrows link the pathological processes to the effects on renal resistance index and renal venous flow. ATII: angiotensin II; CVP: central venous pressure; NO: nitric oxide; RAAS: renin-angiotensin-aldosterone system; ROS: reactive oxygen species; SNA: sympathetic nerve activity.

**Table 2 biomedicines-12-01166-t002:** VExUS grading system. IVC: inferior vena cava. VExUS: venous excess ultrasonography score.

Grade of VExUS	IVC Diameter	Severely Abnormal Wave Pattern	Grade of Venous Congestion
**0**	<2 cm	-	None
**1**	>2 cm	Zero	Mild
**2**	>2 cm	One	Moderate
**3**	>2 cm	Two	Severe

## Data Availability

Not applicable.

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
