# Peer review of "Renal Arterial and Venous Doppler in Cardiorenal Syndrome: Pathophysiological and Clinical Insights"

_biomedicines, 2024, doi:10.3390/biomedicines12061166_

Round 1
Reviewer 1 Report
Comments and Suggestions for Authors
This study is significant in that it aggregates knowledge for clinical use of renal resistive index in renal Doppler as a physiological and pathological index of the kidney. However, during the process up to adoption, the following points need to be added.
1 Introduction
Many papers on the clinical use of RRI have been reported from around 2000 to 2010, and their significance is needed in order to write a review article after 2020. The significance of this review should be clarified by touching on the technological innovations that have occurred since 2019.
2. Figure 2: This figure is considered to have been written as a background for clinically referring to RRI, but there are two problems. ① There is insufficient tracing back to the data that created the diagram. Many reference papers should be added to the legend. The relationship with the description of RRI as a clinical index for each disease/syndrome in ② cannot be seen from this diagram. The relationship between clinical indicators of each disease/syndrome and blood pressure and blood flow velocity in each region should be illustrated.
3. L174-189: Regarding the significance of RRI in predicting diabetes, etc., please refer to the following literature and describe comparisons with indicators other than RRI.
Hussain, Salman, et al. "Potential biomarkers for early detection of diabetic kidney disease." Diabetes Research and Clinical Practice 161 (2020): 108082.
4. L190-208 Regarding the significance of RRI in predicting heart failure, please refer to the following literature and describe comparisons with indicators other than RRI.
EHR
Chen, Robert, et al. "Recurrent neural networks for early detection of heart failure from longitudinal electronic health record data: implications for temporal modeling with respect to time before diagnosis, data density, data quantity, and data type." Circulation: Cardiovascular Quality and Outcomes 12.10 (2019): e005114.
Various imaging data
Pellicori, Pierpaolo, et al. "Ultrasound imaging of congestion in heart failure: examinations beyond the heart." European journal of heart failure 23.5 (2021): 703-712.
5. L210-230 Regarding the significance of RRI in predicting AKI, please refer to the following literature and describe comparisons with indicators other than RRI.
Cui, Hao, et al. "Plasma Metabolites–Based Prediction in Cardiac Surgery–Associated Acute Kidney Injury." Journal of the American Heart Association 10.22 (2021): e021825.
6. L251-268 Regarding COVID-19 infection, please refer to the following documents and demonstrate the significance of RRI measurement.
Adapa, Sreedhar, et al. "COVID-19 and renal failure: challenges in the delivery of renal replacement therapy." Journal of clinical medicine research 12.5 (2020): 276.
7. L298-313 Regarding chronic heart failure, indicators other than RRI are also used, such as the following literature. Please add the significance of RRI in comparison to these or as a combined indicator.
Ter Maaten, Jozine M., et al. "The effect of decongestion on intrarenal venous flow patterns in patients with acute heart failure." Journal of Cardiac Failure 27.1 (2021): 29-34.
Author Response
We thank the reviewer for her/his helpful comment. This is our point-to-point reply:
1 Introduction
Many papers on the clinical use of RRI have been reported from around 2000 to 2010, and their significance is needed in order to write a review article after 2020. The significance of this review should be clarified by touching on the technological innovations that have occurred since 2019.
Response:
Thank you for the comment. We modified the text by adding a paragraph, page 1-2, lines 37-47:
“Technological advances are significantly shaping medical practice. In recent years, machine learning and big data have emerged as valuable tools for identifying high-risk populations and assisting physicians in clinical practice. Within this framework, it is essential to assess whether clinical and instrumental indices, such as RRI, offer comparable utility to these new technological instruments or provide additional value to the data being analyzed [6]. Moreover, RRI has recently been recognized as an indicator of how nephroprotective drugs may affect the kidney. The sodium-glucose cotransporter two inhibitors (SGLT2i) and angiotensin-converting enzyme inhibitor (ACE-I) lead to an in-itial decrease in glomerular filtration rate (GFR), inhibition of oxidative stress markers, additional reductions in blood pressure, and overall peripheral resistance. Notably, long-term gliflozin preserves renal perfusion and lowers vascular resistance [7].”
- Figure 2: This figure is considered to have been written as a background for clinically referring to RRI, but there are two problems. ① There is insufficient tracing back to the data that created the diagram. Many reference papers should be added to the legend. The relationship with the description of RRI as a clinical index for each disease/syndrome in ② cannot be seen from this diagram. The relationship between clinical indicators of each disease/syndrome and blood pressure and blood flow velocity in each region should be illustrated.
Response:
Thank you for the comment. The figure would only summarize the pathophysiological background or arterial and venous Doppler. For this reason, we simplified the figure.
- L174-189: Regarding the significance of RRI in predicting diabetes, etc., please refer to the following literature and describe comparisons with indicators other than RRI. Hussain, Salman, et al. "Potential biomarkers for early detection of diabetic kidney disease." Diabetes Research and Clinical Practice 161 (2020): 108082.
Response:
Thank you for the comment. We added a paragraph (page 5, lines 195-204) relative to the suggested reference which was cited as ref #36:
“Microalbuminuria and serum creatinine have been regarded as predictors of DN. However, their predictive value is limited as their levels rise significantly when renal damage has occurred. Over the last years, alternative biomarkers have been studied to identify early signs of DN. Galectin-3 has been linked to glomerular injury, but high levels have also been observed in the general population, including individuals without CKD. Growth differentiation factor-15 (GDF15) has been implicated in the progression of kidney dysfunction, mortality, and morbidity among patients with type-1 diabetes mellitus and altered kidney function. Nevertheless, there is currently insufficient evi-dence regarding the utility of galectin-3 and GDF-15 as biomarkers for DN [36].”
- L190-208 Regarding the significance of RRI in predicting heart failure, please refer to the following literature and describe comparisons with indicators other than RRI.
HER
Chen, Robert, et al. "Recurrent neural networks for early detection of heart failure from longitudinal electronic health record data: implications for temporal modeling with respect to time before diagnosis, data density, data quantity, and data type." Circulation: Cardiovascular Quality and Outcomes 12.10 (2019): e005114.
Various imaging data Pellicori, Pierpaolo, et al. "Ultrasound imaging of congestion in heart failure: examinations beyond the heart." European journal of heart failure 23.5 (2021): 703-712.
Response
Thank you for the comment. We modified the text adding the suggested references:
Chen et al. as ref # 43, at page 6, lines 225-230:
“Studies have explored the use of kidney and venous system imaging to detect congestion, offering improved prediction of heart failure decompensation [43]. However, these methods may not fully assess the functional residual capacity of the kidneys, even in cases of compensated heart failure. Consequently, the advantage of utilizing the renal resistive index diverges slightly from that of venous congestion imaging, as it reflects the degree of congestion and the renal exertion to sustain diuresis.”
Pellicori et al. as ref # 47, at page 6, lines 244-250:
“In recent years, various machine learning models have been extensively tested for the early detection of heart failure (HF) patients, a subject investigated notably by Chen et al. Among these models, recurrent neural networks (RNNs) have proven notably effective in predicting future heart failure diagnoses, exceptionally when trained on ample datasets [43]. There needs to be more comparative data between these methods and the RRI in predicting CS. Nevertheless, including RRI data within the machine learning model could enhance the accuracy of heart failure diagnoses in this context [47].”
- L210-230 Regarding the significance of RRI in predicting AKI, please refer to the following literature and describe comparisons with indicators other than RRI. Cui, Hao, et al. "Plasma Metabolites–Based Prediction in Cardiac Surgery–Associated Acute Kidney Injury." Journal of the American Heart Association 10.22 (2021): e021825.
Response:
We added the suggested reference as ref #54 and the relative paragraph at page 7, lines 266-280
“Serum creatinine levels have been considered a gold standard for the diagnosis of AKI; however, hit concentrations may be affected by age, sex, muscle mass, and medi-cations, and abnormal levels may not be detectable until patients suffer a significant loss of kidney function. In recent years, novel biomarkers, such as NGAL (neutrophil gelatinase-associated lipocalin), cystatin C, and KIM‐1 (kidney injury molecule‐1), have been associated with renal tubular injury. Although these biomarkers may be helpful, alone or in combination, for improving clinical diagnosis, the sensitivity, specificity, and repeatability remain insufficient for clinical implementation. Serum creatinine levels have traditionally been used as the primary measure for diagnosing AKI; however, several factors can influence their accuracy [53]. Additionally, abnormal levels may not become apparent until a significant decline in kidney function occurs. Recently, new biomarkers like NGAL (neutrophil gelatinase-associated lipocalin), cystatin C, and KIM‐1 (kidney injury molecule‐1) have emerged as indicators of renal tubular injury. Even if these biomarkers can enhance clinical diagnosis, their sensitivity, specificity, and reliability are currently insufficient for widespread clinical application [54].”
- L251-268 Regarding COVID-19 infection, please refer to the following documents and demonstrate the significance of RRI measurement.
Adapa, Sreedhar, et al. "COVID-19 and renal failure: challenges in the delivery of renal replacement therapy." Journal of clinical medicine research 12.5 (2020): 276.
Response:
Thank you for the comment.
We added ref #... and the relative paragraph at page 8, lines 321-330:
“During COVID-19 infection, certain individuals may develop severe complications such as viral pneumonia, multiorgan failure including respiratory failure due to acute res-piratory distress syndrome (ARDS), myocarditis, advanced renal failure, and even death. The occurrence of acute kidney injury (AKI) among COVID-19 patients ranges from 3% to 15%, but in cases requiring intensive care, this rate significantly rises from 15% to 50%. AKI has been identified as an independent risk factor for mortality in individuals with COVID-19. Moreover, COVID-19 infection is linked to an elevated frequency of albu-minuria and hematuria. Since AKI independently contributes to increased hospital mortality among COVID-19 patients, a novel marker (other than serum creatinine) is needed in order to prevent renal damage [67].”
- L298-313 Regarding chronic heart failure, indicators other than RRI are also used, such as the following literature. Please add the significance of RRI in comparison to these or as a combined indicator. Ter Maaten, Jozine M., et al. "The effect of decongestion on intrarenal venous flow patterns in patients with acute heart failure." Journal of Cardiac Failure 27.1 (2021): 29-34.
Response:
Thank you for the comment. We modified the text at page 10, lines 381-385 and added the suggested reference (ref #72):
“Evaluation of intrarenal flow patterns through renal Doppler ultrasonography has garnered attention in patients with heart failure. Discontinuous flow has been proposed to correlate with congestion and diminished diuretic efficacy. Simultaneous recording of pulsed Doppler waveforms from interlobar arteries and veins enables rapid assessment of patient filling status and facilitates monitoring of diuretic response. [72]”
Reviewer 2 Report
Comments and Suggestions for Authors
The topic is very important and the review is useful. I think VEXUS is a very importnat method for identification and follow up of the abdominal (here renal) blood supply. I would like to ask you to add the following parts to the paper:
-schematic presentation , where the probe should be placed in sittind and also in lying position
-application in intensive care settings including the status of the transrenal flow, classification
still ICU detection of abdominal congestion (with pics) and fluid overload
role of intravenous vasopressores in the renal flow (like norepinephrine and vasopressine)
-please refer the existing ICU and emergency care ultrasound methods
please use same metrics on the renal doppler pics (y axis) and give schematic description how to interpret them
Comments on the Quality of English Languageit is ok
Author Response
We would like to thank the reviewer for his/her helpful comments. This is our point-to-point reply.
The topic is very important and the review is useful. I think VEXUS is a very important method for identification and follow up of the abdominal (here renal) blood supply. I would like to ask you to add the following parts to the paper.
- schematic presentation, where the probe should be placed in sitting and also in lying position
Response:
Thank you for the comment. We modified the text to better explain this topic, at page 9, lines 369-373:
“As for the arterial Doppler, low-frequency probe (2.5-5.0 MHz) visualizes the kidney and its vessels. Images can be taken through two primary approaches: with patient in sitting position or in lying position, placing the probe on the flank on the right posterior axillary line. Doppler waveforms are obtained on interlobar and segmental renal veins. Because the venous signal is very dependent on the respiratory phase, Doppler spectra are ob-tained at the end of expiration. Doppler waveforms of intrarenal veins are classified as in figure 1 (from continuous to intermittent pattern). Arterial flow will show a positive tracing, while venous flow will show a negative tracing.”
-role of intravenous vasopressores in the renal flow (like norepinephrine and vasopressine)
Response:
Thank you for the comment. It is very relevant but it is beyond the aim of our review.
-application in intensive care settings including the status of the transrenal flow, classification
-still ICU detection of abdominal congestion (with pics) and fluid overload
-please refer the existing ICU and emergency care ultrasound methods
Response:
Thank you for the comment. We added a section about the evaluation of transrenal flow in ICU at page 9-10, lines 409-438:
“An accurate assessment of fluid status is crucial for effectively treating patients in the emergency department. Traditionally, clinicians have relied on examining the inferior vena cava (IVC) and detecting pulmonary B-lines to gauge fluid status and responsiveness [77].
The Venous Excess Ultrasonography Score (VExUS) offers a comprehensive approach. It involves a four-step process that not only checks for congestion in the IVC but also evaluates the severity of congestion in the liver, gut, and kidneys. This protocol equips emergency physicians with a tool to tailor fluid management strategies for their patients, including decisions on fluid administration, diuretic use, and vasopressor therapy, and could be used as part of Protocols for Point-of-Care-Ultrasound (POCUS) estimation [78].
The assessment begins by examining the diameter of the inferior vena cava (IVC). If the IVC diameter measures less than 2 cm, congestion is absent, and further evaluation is unnecessary (VExUS 0). However, if the diameter exceeds 2 cm, congestion is present, and further investigation to determine its severity are needed. Next, the hepatic veins are assessed, with any of the three (right, middle, and left) suitable for evaluation. Using color Doppler, the veins should appear blue. The pulsed wave Doppler is positioned proximal to the hepatic vein's entry into the IVC. Normal hepatic vein flow exhibits a trace pattern with three waves: a small retrograde A wave, followed by anterograde S and D waves. In cases of venous congestion, the S wave's magnitude diminishes, eventually becoming positive, signifying a reversal of blood flow during systolic phase venous flow reduction. Following the hepatic vein assessment, attention turns to the portal vein, identified by its thick and hyperechoic walls. Using color Doppler, the portal veins should appear red. The pulsed wave Doppler is placed in the hepatic portal vein. In normal conditions, the portal system exhibits constant monophasic flow with minimal variation. However, as venous congestion increases, the flow becomes pulsatile. The final step involves evaluating of the renal vein, following the same methodology as described earlier. Normal renal veins display uninterrupted monophasic flow, but as venous congestion escalates, the systolic component of the flow decreases [79].
The VExUS score categorizes congestion into Grades 0-3 as shown in Table 2.”
- please use same metrics on the renal doppler pics (y axis) and give schematic description how to interpret them
Response:
Thank you for the comment. We tried to better describe it in the text.
Reviewer 3 Report
Comments and Suggestions for Authors
This manuscript gives an overview of the significance of renal Doppler measurements, specifically the renal resistive index (RRI), in the context of cardiovascular and renal diseases. However it lacks any mention of the methodology of the review. Including a brief note on how the data were collected or selected (e.g., types of studies reviewed, databases searched) could improve the reader's understanding of the review's scope and rigor. A statement summarizing the impact of these findings on clinical practice or future research would also be valuable. The abstract is well-written, but the hyphenation in "usefulness" and "associated" is unusual and may result from formatting issues. Ensure that the text is correctly formatted to avoid potential confusion.
Author Response
This manuscript gives an overview of the significance of renal Doppler measurements, specifically the renal resistive index (RRI), in the context of cardiovascular and renal diseases.
We would like to thank the reviewer for his/her helpful comments. This is our point-to-point reply.
- However, it lacks any mention of the methodology of the review. Including a brief note on how the data were collected or selected (e.g., types of studies reviewed, databases searched) could improve the reader's understanding of the review's scope and rigor.
Response:
Thank you for the comment. We modified the text to better explain the topic: from line 48 to line 56.
“This review offers perspectives on the clinical utility of assessing arterial and venous renal Doppler in conditions affecting both the heart and kidneys. In order to achieve the goal of our research, we conducted searches in electronic databases such as PubMed, using specific keywords or locating articles related to renal resistance index and cardiovascular diseases, arterial renal Doppler, venous renal Doppler and congestion, and cardiorenal syndrome. Following the primary keyword searches in databases and adhering to the predetermined criteria for including and excluding literature, we selected studies based on the incidence rate of listening issues and the frequency of pure tone hearing threshold studies.”
- A statement summarizing the impact of these findings on clinical practice or future research would also be valuable.
Respose:
Thank you for the comment. We modified the text to better explain some topics. From line 389 to line 392:
“These discoveries indicate the practical value of assessing renal venous Doppler in gaining deeper insights into cardio-renal syndrome. Incorporating renal venous Doppler assessments into routine clinical practice could be valuable, allowing for the customization of treatment strategies accordingly.”
- The abstract is well-written, but the hyphenation in "usefulness" and "associated" is unusual and may result from formatting issues. Ensure that the text is correctly formatted to avoid potential confusion.
Response:
Thank you for the comment.
We modified the abstract as follows:
“In recent decades, considerable effort has been put into investigating renal Doppler measurements' clinical utility in cardiovascular and renal disorders. In particular, a measure of renal arterial resistance, the renal resistive index (RRI), has been demonstrated to predict chronic kidney disease progression and acute kidney injury in different clinical settings. Furthermore, it is linked to a poorer prognosis in individuals suffering from chronic heart failure. Examining renal venous flow through pulsed Doppler can offer additional insights into renal congestion and cardiovascular outcomes for these patients. This review seeks to summarize the existing data concerning the clinical significance of arterial and venous renal Doppler across various cardiovascular and renal disease contexts.”
Round 2
Reviewer 2 Report
Comments and Suggestions for Authors
The authors subitted a revised version of the manuscript. My major concerns are the following:
1. It would be still useful to schematically show the place of the ultrasound probe on a body torso. Second, the ultrasound pictures should be also used showing the RRI calculations and the explanation of the changes by a diagram
2. The Table 1. should contain the major findings of the listed studies with numbers not only a short description (like outcome variables, main findings)
3. In acute settings, the kidney ultrasound has important role. So we cannot replace it with the portal vein investigations. In this section, the renal flow of VEXUS should be discussed and not the portal and hepatic flow methods.
4. PLease add numbers to your statements , as only qualitative statement does not help to the readers.
I would like to ask the authors to add practical hints (like the placement of probe in a graph) and calculation methods (by explaining the ultrasound differences between the normal and abnormal stages, like new waves, etc)
and using more specified recommendations.
Comments on the Quality of English Languageit seems fine
Author Response
We thank the reviewer for her/his helpful comment. This is our point-to-point reply
- It would be still useful to schematically show the place of the ultrasound probe on a body torso. Second, the ultrasound pictures should be also used showing the RRI calculations and the explanation of the changes by a diagram
Response:
We changed figure 1 by creating two new figures.
The new Figure 1 shows:
- the position of patient and of the probe
- The calculation of renal resistance index
- The examples showing the increase of resistance index and the factors influencing its calculation
The new Figure 2 shows:
- The evaluation of renal venous Doppler
- The examples showing the reduction of renal venous flow and the Doppler patterns associated to it.
- The Table 1. should contain the major findings of the listed studies with numbers not only a short description (like outcome variables, main findings)
Response:
We modified the table as suggested.
- In acute settings, the kidney ultrasound has important role. So we cannot replace it with the portal vein investigations. In this section, the renal flow of VEXUS should be discussed and not the portal and hepatic flow methods.
Response:
We've discussed the portal and hepatic flow methods as they're part of the VExUS assessment. However, we've revised the text, minimizing the discussion on portal and hepatic veins and providing a clearer explanation of renal flow, at page 11-12, lines: 431-445:
The assessment begins by examining the diameter of the inferior vena cava (IVC). If the IVC diameter measures less than 2 cm, congestion is absent, and further evaluation is unnecessary (VExUS 0). However, if the diameter exceeds 2 cm, congestion is present, and further investigation to determine its severity are needed. Next, the hepatic veins are assessed, with any of the three (right, middle, and left) suitable for evaluation. Using color Doppler, the veins should appear blue. The pulsed wave Doppler is positioned proximal to the hepatic vein's entry into the IVC. Normal hepatic vein flow exhibits a trace pattern with three waves: a small retrograde A wave, followed by anterograde S and D waves. In cases of venous congestion, the S wave's magnitude diminishes, eventually becoming positive, signifying a reversal of blood flow during systolic phase venous flow reduction. Following the hepatic vein assessment, attention turns to the portal vein, identified by its thick and hyperechoic walls. Using color Doppler, the portal veins should appear red. The pulsed wave Doppler is placed in the hepatic portal vein. In normal conditions, the portal system exhibits constant monophasic flow with minimal variation. However, as venous congestion increases, the flow becomes pulsatile. The final step involves evaluating of the renal vein, following the same methodology as described earlier. Normal renal veins display uninterrupted monophasic flow or slightly pulsatile, but as venous congestion increases, intrarenal venous flow shows gradual changes, from a biphasic pattern in moderate congestion to a monophasic pattern in severe congestion [79].
- Please add numbers to your statements, as only qualitative statement does not help to the readers.
I would like to ask the authors to add practical hints (like the placement of probe in a graph) and calculation methods (by explaining the ultrasound differences between the normal and abnormal stages, like new waves, etc) and using more specified recommendations
Response:
In Figure 1 and in the table 1 we added more data useful for performing and interpreting arterial renal Doppler.
Round 3
Reviewer 2 Report
Comments and Suggestions for Authors
I am satisfied with your responses. Thak you for your extra efforts. More practical. and remains comprehensive.